# Effect of digitizing Community Health Information System on the delivery and use of maternal and child health services: Propensity score matching analysis

Gizachew Tadele Tiruneh[1]*, Girma Tadesse[1], Alemnesh Mirkuzie Hailemariam[1], Yared Kifle[1], Tsegaye Shewangzaw[1], Bezawit Mesfin Hunegnaw[1], Getnet Alem[1], Nebreed Fesseha[1], Biruk Bogale[1], Gete Mekuria[2], Alemayehu Hunduma[3], Dessalew Emaway[1]

1 JSI, Addis Ababa, Ethiopia, 2 Children's Investment Fund Foundation, Addis Ababa, Ethiopia, 3 Ministry of Health, Addis Ababa, Ethiopia

* gizt121@gmail.com

## Abstract

### Introduction

In 2018, Ethiopia's government digitized its Community Health Information System (eCHIS) to enhance Health Extension Program service delivery. However, the impact of eCHIS on health outcomes remains unclear. This study, therefore, examined the effects of eCHIS on the uptake of maternal and child health services.

### Methods

A post-test-only, non-equivalent group household survey design was used, collecting data from July to August 2024 in rural communities with and without eCHIS. A stratified multistage sampling technique was employed to recruit respondents. A sample of 1,728 women of reproductive age (271 in eCHIS and 1,457 in non-eCHIS woredas), 1,118 women with children ages 0–11 months (188 in eCHIS and 930 in non-eCHIS), and 569 women with children ages 12–23 months (301 in eCHIS and 268 in non-eCHIS) were included in this study.
Propensity scores were used to match intervention and comparison communities based on women's age, religion, parity, education, household wealth, distance to health facilities, and autonomy. A modified Poisson regression analysis, adjusting for covariates, was conducted to estimate the adjusted prevalence ratios (PR) for maternal and child health outcomes.

### Results

Modern contraceptive use was 66% in intervention areas versus 56% in comparison areas. Institutional delivery rates were 95% in intervention and 79% in comparison

**Data availability statement:** The dataset used and analyzed during this study is included as supplementary information to this article (S2 File).

**Funding:** Bill and Melinda Gates Foundation, IVN-057145 (to D.E.), and Children's Investment Fund Foundation, R-2010-05155 (to D.E.).

**Competing interests:** The authors have declared that no competing interests exist.

**Abbreviations:** ANC, antenatal care; CHIS, Community Health Information System; CIFF, Children's Investment Fund Foundation; CPR, contraceptive prevalence rate; eCHIS, electronic Community Health Information System; HEP, Health Extension Program; HEW, Health Extension Worker; mHealth, mobile health; MNH, maternal and newborn health; PNC, post-natal care; PR, prevalence ratio; RMNCH, reproductive, maternal, newborn, and child health; SD, standard deviation; SSD, Strengthening Service Delivery in Ethiopia.

areas. In intervention areas, 94% of children received the first dose of the Pentavalent vaccine compared to 83% in comparison areas, while 85% and 68%, respectively, received the third dose. Adjusted analyses showed that eCHIS intervention areas had statistically significant increases in contraceptive use (13.0 percentage points; PR: 1.25; p-value < 0.01) and institutional deliveries (8.9 percentage points; PR: 1.18; p-value < 0.01) compared to comparison areas. Similarly, the average treatment effect of eCHIS on Pentavalent 1, Pentavalent 3, and full vaccination coverage showed significant increases of 10.3 (PR: 1.13; p-value < 0.01), 13.6 (PR: 1.26; p-value < 0.01), and 14.6 (PR: 1.36; p-value < 0.01) percentage points, respectively.

## Conclusion

This study underscores the transformative potential of eCHIS in improving maternal and child health outcomes in Ethiopia. Sustained investment in digital health systems can help scale effective health care services to rural and underserved communities.

## Background

Since 2003, the Health Extension Program (HEP) in Ethiopia has aimed to improve access to health services in rural communities by deploying trained Health Extension Workers (HEWs) to provide essential health services at the community level. The program has significantly increased access to maternal and child health services, particularly in rural and underserved areas [1,2], and has contributed to a decline in maternal and child mortality rates in Ethiopia [3,4].

Despite its successes in expanding access to essential health services, the HEP faces several challenges, including inadequate infrastructure, shortages of supplies and human resources, and a lack of quality data for program management and monitoring [5]. In 2008, Ethiopia's Ministry of Health (MOH) introduced a paper-based Community Health Information System (CHIS) to track HEP implementation. To further enhance program effectiveness, MOH launched the electronic Community Health Information System (eCHIS) in 2018, a digital platform designed to improve community-level health care delivery and support data-informed decision-making at all levels of the health care system. The eCHIS digitizes data collection for the 18 HEP service packages, including immunization, maternal and child health, and disease management, supporting HEWs in delivering more efficient and effective services [6].

The eCHIS system includes three mobile applications: the HEW app for managing health services, the Focal Person app for monitoring and support, and a Referral app for coordinating referrals to and from health centers [7]. The HEW app digitized family folder which maintains detailed household records, including maternal and child health information, allowing HEWs to identify eligible women for reproductive, maternal, newborn, and child health (RMNCH) services and effectively monitor and manage HEP services. Key features of eCHIS—including family planning, maternal and child health, household registration, appointments, referrals, and job aids—enhance

service delivery and uptake by facilitating referrals and tracking defaulters. The appointment feature enables HEWs to prepare for RMNCH services at health posts or during home visits, while the referral system directs clients to nearby health centers for appropriate care, including institutional deliveries. Immunization tracking ensures timely vaccinations, supports ongoing immunization efforts, and reduces defaulters by maintaining children's immunization histories and scheduling vaccinations. Additionally, job aids offer HEWs quick guidance for consultations and targeted service delivery [6].

Previous studies have shown that community-based data and CHIS increased local data use by community volunteers and HEWs [8,9]. The implementation of eCHIS has improved health data accuracy and enabled better tracking of antenatal care visits, immunizations, and child health outcomes. Additionally, eCHIS supports timely decision-making by health workers, enhances service coverage by identifying gaps in underserved areas, and promotes efficient service delivery, contributing to increased uptake of RMNCH services [7,10–12]. However, significant knowledge gaps remain regarding eCHIS's impact on health outcomes [7], which need to be addressed. Therefore, this study examined the effect of eCHIS on the delivery and uptake of RMNCH services.

## Methods

### Setting

In Ethiopia, the woreda-based primary health system typically comprises one primary hospital, four to five health centers, and 20–25 health posts, collectively serving a population of approximately 100,000 people. This primary health care network provides a range of services—including promotive, preventive, curative, and rehabilitative care—through health facilities (health posts, health centers, and primary hospitals), at homes and in communities, and via outreach efforts as part of the HEP.

Since December 2020, JSI has been implementing the "Transforming Community Health-Electronic Community Health Information System (eCHIS)" project, funded by the Children's Investment Fund Foundation (CIFF), to support the MOH's efforts to scale up eCHIS nationwide. The project also aims to strengthen the HEP by enhancing data utilization for effective performance management of HEWs, ultimately improving health outcomes for mothers and children.

The project employs a phased implementation approach to scale up eCHIS across 110 agrarian woredas. The initial phase covered 20 woredas from December 2020 to November 2021, followed by an additional 44 woredas during the December 2022–November 2023 fiscal year. The remaining woredas are scheduled to begin eCHIS implementation after December 2023.

A household survey was conducted to establish baselines for monitoring the impact of the *"Strengthening Service Delivery in Ethiopia (SSD)"* project on RMNCH outcomes. JSI was awarded this project by the Gates Foundation in November 2023 to be implemented across 20 woredas in seven regions of Ethiopia: Afar, Amhara, Oromia, Sidama, Central Ethiopia, Somali, and Southwest Ethiopia. The baseline survey covered 14 of the 16 agrarian SSD intervention woredas (including five eCHIS and nine non-eCHIS) and all four pastoral woredas. Of the surveyed eCHIS woredas, four began implementation in the first phase, while one started after December 2022.

### Design

This study used the baseline household survey dataset collected primarily to monitor RMNCH service utilization for the SSD project between July and August 2024. A post-test-only non-equivalent group design was nested within the household survey to evaluate the effect of eCHIS on RMNCH service utilization by comparing eCHIS-implementing and non-eCHIS woredas.

### Sample size and data collection

The SSD baseline survey used a stratified, multistage sampling approach by program domains and agrarian-pastoral regions to recruit three groups of women aged 15–49 years: 3,611 women of reproductive age, 1,922 women with infants

aged 0–11 months, and 1,368 women with children aged 12–23 months. The household selection used a two-stage stratified cluster sampling method. In the first stage, 30 kebeles (i.e., the smallest administrative unit) per program domain were randomly chosen in agrarian areas, while all 50 kebeles were included in pastoral regions, stratified by program domain. Sample size per kebele was proportionally allocated based on population size. Household lists of eligible women were obtained from each kebele's family folder, serving as sampling frames. In the second stage, households were systematically sampled proportionate to kebele population size (Fig 1).

Since eCHIS has not been implemented in pastoral settings, this study focused on 90 agrarian kebeles (47 eCHIS and 43 non-eCHIS). The final sample sizes, therefore, included 1,728 women of reproductive age (271 eCHIS and 1,457 non-eCHIS), 1,118 women with children ages 0–11 months (188 eCHIS and 930 non-eCHIS), and 569 women with children ages 12–23 months (301 eCHIS and 268 non-eCHIS).

The survey gathered family planning information from women of reproductive age, maternal and newborn health (MNH) care data from women who had given birth in the 12 months prior, and information on child immunization and common childhood illnesses from women with children ages 12–23 months. Questionnaires (S1 File) were administered in Amharic

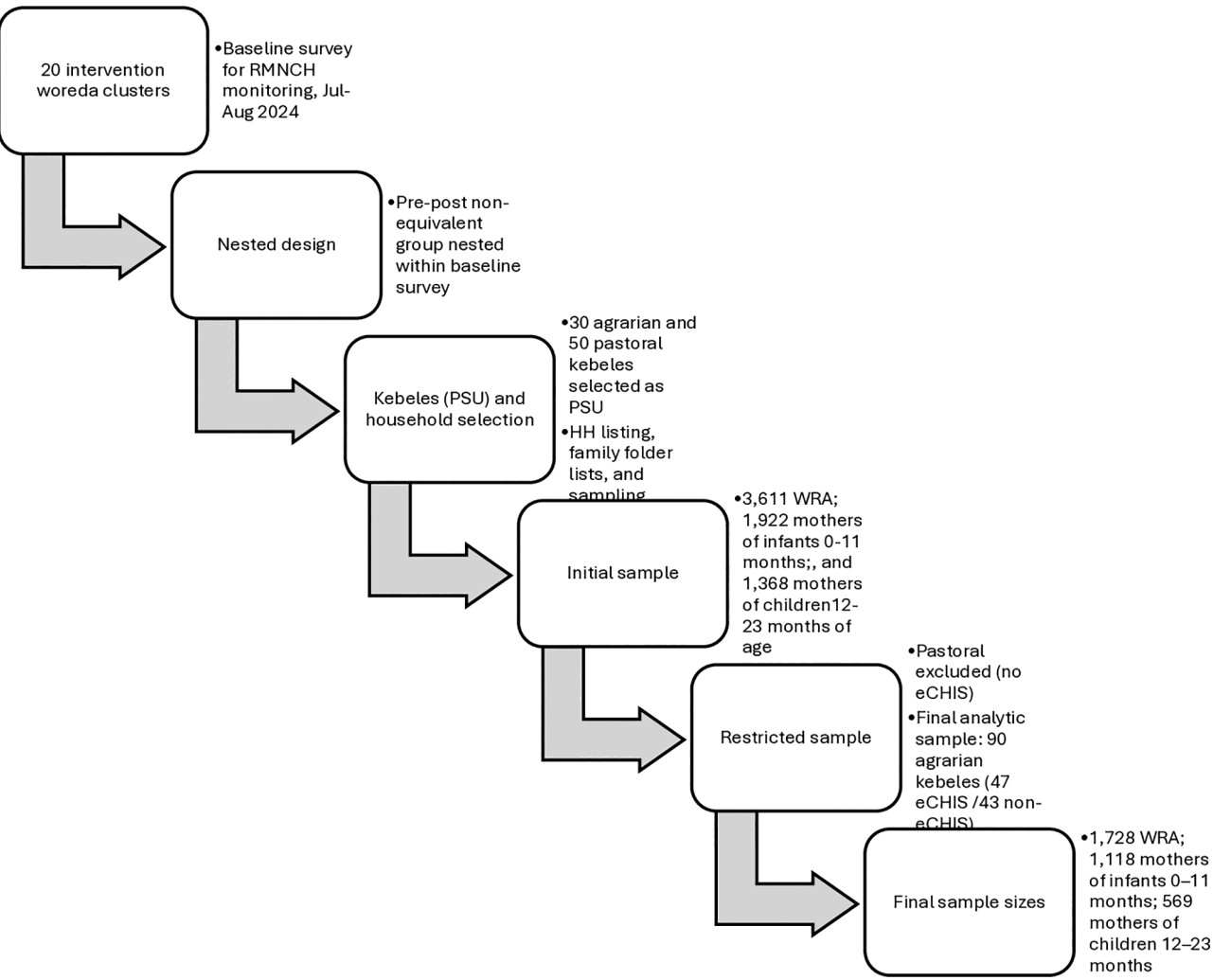

**Fig 1. Sampling design and survey process.**

for the Afar, Amhara, Central Ethiopia, Sidama, and Southwest Ethiopia regions, in Af-Somali for the Somali region, and in Oromiffa for the Oromia region, with data collection facilitated through a web-based mHealth platform (SurveyCTO) using tablets and smartphones. The data collectors received a 4-day theoretical and technical training on the questionnaire and data collection tool with one-day field pretesting of the questionnaire.

## Measurements

The dependent variable of interest was the RMNCH service uptake, which was expected to be affected by the eCHIS intervention and was measured by the household surveys among three groups of women aged 15–49 years: women of reproductive age for family planning indicators, women with infants aged 0–11 months for MNH indicators, and women with children aged 12–23 months for immunization indicators. The variables were calculated from the sampled target/denominator to create indicators. Table 1 below presented the indicators computed for this paper, along with their definitions and data sources.

The independent variables that were considered as potential confounders were maternal age, maternal education, marital status, parity, religion, household wealth, and distance of the respondents' household from the nearest health facility (either health post, health center, or hospital). A wealth index score was constructed for each household using Principal Component Analysis (PCA) based on the household's possessions (i.e., bank account, electricity, watch, radio, television, mobile phone, telephone, refrigerator, bed, electric stove, bicycle, motorcycle, and three-wheeler bajaj), assets (including number of goats, milk cows or bulls, sheep, horses, donkeys, mules, camels, and chickens), and characteristics (such as type of latrine, water source, floor, and wall material). The first principal component, which captured the largest variation in asset ownership and living conditions and had the highest eigenvalue, was retained to represent underlying socioeconomic status and used to generate a continuous wealth score for each household. Households were then ranked according to their wealth scores and divided into three quintiles [13]. Women's autonomy was constructed from three items: women's sole or joint decision-making regarding the use of earned money, health care for themselves, and household purchases. A woman was categorized as having autonomy if she fulfilled at least two of the three criteria; otherwise, she was labeled as having no autonomy.

## Analysis

Data were analyzed using Stata version 15.1, applying weighted samples where appropriate. Pearson's chi-squared test was used to compare the background characteristics of respondents and unadjusted RMNCH indicators between eCHIS and non-eCHIS intervention groups adjusted for cluster survey design effects.

Propensity scores were calculated for each kebele to estimate adjusted intervention effects, using a logit model for the three groups of mothers separately, accounting for the clustered sampling design in which households within the same kebele share similar contextual characteristics and reducing bias from correlated exposures. Covariates with a p-value greater than 0.2 in the logit model were excluded using a stepwise-backward selection procedure, which was conducted separately for each subgroup [4,5].

Accordingly,

1) For women of reproductive age, the final logit model included the following covariates: women's age, parity, education, household wealth, distance to a health facility, and women's autonomy.

2) For women with children ages 0–11 months, the covariates were women's age, religion, parity, education, household wealth, and distance to a health facility.

3) For women with children ages 12–23 months, the covariates included religion, education, distance to a health facility, and women's autonomy.

**Table 1. Definition of RMNCH indicators and their data sources.**

| Indicators | Definitions | Data sources |
|---|---|---|
| Modern method contraceptive prevalence rate (CPR) | Percentage of married women of reproductive age who were using modern contraceptive methods. | Interview with women of reproductive age group |
| Antenatal care (ANC) 1 or more contacts | Percentage of women who received at least one ANC contact with a health professional (doctor, nurse, midwife, or HEWs) during their last pregnancy | Interview with women of reproductive age group with 0–11 months of age children |
| ANC 4 or more contacts | Percentage of women who received at least four ANC contacts with a health professional (doctor, nurse, midwife, or HEWs) during their last pregnancy | Interview with women of reproductive age group with 0–11 months of age children |
| Institutional delivery | Percentage of women who delivered at a health facility (health center, hospital, or private clinic) during their last childbirth | Interview with women of reproductive age group with 0–11 months of age children |
| Postnatal care (PNC) within 6 weeks | Percentage of women who received at least one postnatal check-up within six weeks post-partum after their most recent childbirth. | Interview with women of reproductive age group with 0–11 months of age children |
| Pentavalent 1 | Percentage of women with children ages 12–23 months who reported their child had received the first dose of the Pentavalent vaccine. | Interview with women of reproductive age group with 12–23 months of age children |
| Pentavalent 3 | Percentage of women with children ages 12–23 months who reported their child had received the third dose of the Pentavalent vaccine. | Interview with women of reproductive age group with 12–23 months of age children |
| All basic vaccinations | Percentage of women with children ages 12–23 months whose children had received the BCG vaccine, three doses of Polio and Pentavalent vaccines, and the measles vaccine (MCV1) | Interview with women of reproductive age group with 12–23 months of age children |

The adequacy of matching was assessed using Stata's "*tebalance*" post-estimation command to ensure that the covariates in the final logit model were not statistically significant (p > 0.1) between intervention and control areas after matching. Finally, the intervention effects were estimated using the *"psmatch2"* command following the matching of the intervention and comparison groups. Subsequently, we conducted a modified Poisson regression model with a log link function and robust variance, to estimate the adjusted prevalence ratios (PR) for RMNCH outcomes with 95% confidence intervals (CI), while controlling for covariates.

## Ethics

Ethical clearance for the study was obtained from the Ethiopian Public Health Association's Research Ethics Review Committee (Ref. #: EPHA/OG/382/24, June 28, 2024). Written permission to conduct the study was sought from the regional health bureaus, zonal health departments, and relevant institutions.

Informed consent was obtained from all participants after providing them with comprehensive information about the research to ensure they fully understood the study. The consent form was approved by the IRB (Ref. #: EPHA/OG/382/24, June 28, 2024), and voluntary participation was ensured throughout the study. All participants were informed about the study's purpose, benefits, and potential risks, and they were made aware of their right to opt out or decline to answer

any questions. When a respondent agreed to participate after reviewing the consent information, the interviewer documented consent by marking the questionnaire and providing a digital signature below the consent statement. Interviews proceeded only after consent was given and properly documented. The researchers ensured that all information obtained from participants was kept confidential and private to respect their privacy and maintain the integrity of the study.

## Results

### Background characteristics of respondents

The respondents' mean age was 28.6 years (standard deviation [SD] = 5.8). Approximately three-fourths were between 20 and 34 years old. More than one-third of the respondents were illiterate, Muslim, and lived within a 30-minute walking time to the nearest health facility. The sociodemographic characteristics of respondents did not significantly differ between the intervention (eCHIS) and comparison (non-eCHIS) sites, except for parity (Table 2).

### Unadjusted estimates of RMNCH indicators

In intervention areas, 66% of married women reported current use of modern contraceptives, compared to 56% in comparison areas. The proportion of mothers delivering in health facilities was 95% in intervention areas versus 79% in comparison areas. For childhood immunizations, 94% of children in intervention areas received the first dose of the Pentavalent vaccine, compared to 83% in comparison areas, while 85% and 68%, respectively, received the third dose. Additionally, 70% of children in intervention areas were fully vaccinated, compared to 55% in comparison areas. As shown

**Table 2. Sociodemographic characteristics of respondents, July-August 2024.**

| | Non-eCHIS | eCHIS | Total |
|---|---|---|---|
| **Mean ages (SD)** | 28.6 (5.9) | 28.7 (5.4) | 28.6 (5.8) |
| **Age category** | | | |
| <20 | 3.4 | 1.6 | 2.9 |
| 20-34 | 75.5 | 79.2 | 76.5 |
| 35-49 | 21.2 | 19.2 | 20.7 |
| **Parity** | | | |
| 0-1 | 27.4** | 26.3 | 27.1 |
| 2-3 | 36.0 | 46.7 | 38.8 |
| 4+ | 36.6 | 27.0 | 34.1 |
| **Religion** | | | |
| Christian | 61.4 | 62.5 | 61.7 |
| Muslim | 38.6 | 37.5 | 38.3 |
| **Educational status** | | | |
| Can't read and write | 37.4 | 34.3 | 36.7 |
| Primary | 24.4 | 24.9 | 24.5 |
| Secondary or higher | 38.2 | 40.8 | 38.9 |
| **Walking time to the nearest facility (either health post, health center, or hospital)** | | | |
| <30 min | 39.4 | 36.0 | 38.5 |
| 30-60 min | 38.2 | 33.3 | 36.9 |
| >60 min | 22.4 | 30.7 | 24.6 |

*$p$-value < 0.01; ** $p$-value <0.05.

in the table below, unadjusted estimates indicated statistically significantly higher coverage in intervention woredas for modern contraceptive use, institutional deliveries, and vaccinations than in comparison woredas (Table 3).

**Intervention effects in RMNCH service uptake**

After adjusting for background characteristics, the eCHIS intervention woredas had statistically significantly higher rates of modern contraceptive prevalence rate (CPR) and institutional deliveries, with increases of 13.0 percentage points (prevalence ratio [PR]: 1.25; p-value <0.01) and 8.9 percentage points (PR: 1.18; p-value <0.01), respectively, compared to the comparison woredas. This indicates that the prevalence of contraception and institutional deliveries in eCHIS woredas was 1.25 times (or 25% higher) and 1.18 times (or 18% higher) than in the comparison areas, respectively. Similarly, the average treatment effect of the eCHIS intervention on Pentavalent first dose, Pentavalent third dose, and full vaccination coverage showed statistically significant increases of 10.3 (PR: 1.13; p-value <0.01), 13.6 (PR: 1.26; p-value <0.01), and 14.6 (PR: 1.36; p-value <0.01) percentage points, respectively, compared with the non-eCHIS woredas. This indicates that eCHIS woredas had 13%, 26%, and 36% higher coverage for Pentavalent 1, Pentavalent 3, and full immunization, respectively, than the comparison woredas (Table 4).

## Discussion

This study demonstrates that eCHIS is effective in improving the delivery and use of essential maternal and child health outcomes, with substantial gains in contraceptive use, institutional deliveries, and vaccination coverage among intervention woredas. These findings suggest that digitization of community health information systems can strengthen primary health care performance when effectively implemented on a scale.

The Ministry of Health in Ethiopia has digitized its CHIS to boost the performance of HEWs and enable more targeted health interventions, ultimately improving service delivery and health outcomes [6]. By equipping HEWs with the eCHIS, workflows are streamlined, data collection and patient tracking are more efficient, and decision-making is better informed, facilitating timely, high-quality care [6,12].

The study provided evidence that eCHIS positively influences the delivery and use of RMNCH services in rural communities. Key system functionalities, including household registration, family-level tracking, appointment scheduling, referral management, and embedded job aids, appear to directly contribute to improved service uptake. For example, client registration and tracking modules likely support early identification of pregnant women and continuity of care, while

**Table 3. Maternal and child health service uptake between eCHIS and non-eCHIS woredas, July-August 2024.**

| Indicators | eCHIS | Non-eCHIS |
|---|---|---|
|  | % (95% CI) | % (95% CI) |
| Modern CPR | 66.0 (59.5-72.5) * | 55.9 (53.1-58.8) |
| ANC 1 or more contacts | 93.1 (89.4-96.7) | 89.9 (88.0-91.8) |
| ANC 4 or more contacts | 62.2 (55.2-69.2) | 56.8 (53.6-60.0) |
| Institutional delivery | 94.7 (91.4-97.9) * | 78.6 (76.0-81.2) |
| PNC within 6 weeks | 52.7 (45.5-60.0) | 46.8 (43.6-50.0) |
| Pentavalent 1st dose | 94.4 (91.7-97.0) * | 82.8 (78.3-87.4) |
| Pentavalent 3rd dose | 85.3 (81.4-89.4) * | 68.3 (62.7-73.9) |
| All basic vaccinations | 70.0 (65.2-75.6) * | 54.9 (48.9-60.8) |

*p-value<0.01; ** p-value <0.05.

**Table 4. eCHIS intervention effects in RMNCH service uptake, July-August 2024.**

| Indicators | eCHIS, n (%) | Non-eCHIS, n (%) | ATT | PR (p-value) |
|---|---|---|---|---|
| Modern CPR | 193 (65.8) | 1,172 (52.8) | 13.0 | **1.25 (<0.01)** |
| ANC 1 or more contacts | 179 (92.7) | 892 (91.6) | 1.1 | 1.02 (0.297) |
| ANC 4 or more contacts | 179 (63.1) | 982 (58.7) | 4.5 | 1.09 (0.181) |
| Institutional delivery | 179 (95.5) | 892 (86.6) | 8.9 | **1.18 (<0.01)** |
| PNC within 6 weeks | 179 (53.1) | 892 (54.7) | −1.7 | 1.00 (0.981) |
| Pentavalent 1st dose | 301 (94.4) | 267 (84.1) | 10.3 | **1.13 (<0.01)** |
| Pentavalent 3rd dose | 301 (85.4) | 267 (71.8) | 13.6 | **1.26 (<0.01)** |
| All basic vaccinations | 301 (70.4) | 267 (55.8) | 14.6 | **1.36 (<0.01)** |

appointment and defaulter-tracing features facilitate timely immunization and follow-up. Similarly, electronic decision-support tools and job aids may reinforce adherence to clinical protocols, contributing to improved quality of care [12].

These findings are consistent with prior studies showing that digital health and mHealth interventions enhance the performance of community health workers by improving accountability, workflow efficiency, and data use for decision-making [12,14–19]. In line with broader mHealth literature, such improvements have been associated with increased uptake of maternal and child health services, including antenatal care attendance and timely immunization [12,14–17,20,21].

At the same time, this study contributes new evidence from the Ethiopian context by quantifying the effects of eCHIS on specific RMNCH outcomes using a quasi-experimental design. The observed improvements in contraceptive use, institutional delivery, and vaccination coverage suggest that integrating digital tools into the Health Extension Program can translate system-level efficiencies into measurable population-level gains.

The limited impact of eCHIS on antepartum and postpartum care, despite its capacity to support appointment reminders and follow-up, may be due to low fidelity in eCHIS implementation and challenges such as infrastructure limitations documented in previous studies [10,22]. Additionally, policy directives recommend that first and fourth ANC visits be conducted at health centers and hospitals, which reduces service provision at health posts and often results in HEWs referring clients instead of providing services at the community level. Furthermore, PNC remains deprioritized and poorly implemented in Ethiopia due to sociocultural factors, low community awareness, and challenges in service delivery [23,24].

The eCHIS represents a significant policy advancement aimed at enhancing the community-based primary health care service delivery model in the country. However, maximizing its impact will require addressing implementation bottlenecks, strengthening infrastructure, and aligning with service delivery policies [10].

Using a propensity score matching analysis technique and data from a post-test-only, non-equivalent group household survey, this research examined the effect of eCHIS on maternal and child health service delivery and utilization, addressing significant knowledge gaps regarding its impact on health outcomes. While propensity score matching improves comparability between intervention and comparison groups, it cannot fully account for unobserved confounders, and the absence of baseline data limits causal inference. in a post-test-only evaluation helps mitigate selection bias and improves group comparability by matching participants in the treatment and comparison groups on observable characteristics, this approach has limitations. Unobserved confounders can still bias the estimated treatment effects, and the lack of baseline data makes it challenging to distinguish true treatment effects from external influences, limiting the robustness of causal inference. Additionally, because the study was not originally designed to evaluate eCHIS, baseline imbalances between intervention and comparison groups may persist despite matching. These likely reflect differences in geographic, socioeconomic, and health system contexts across clusters, which could influence both exposure and outcomes. Residual confounding may therefore affect the estimated effects. Furthermore, reliance on 12-month recall of self-reported behaviors introduces potential recall and social desirability biases.

## Conclusions

In conclusion, this study highlights the transformative potential of Ethiopia's digitized CHIS in advancing maternal and child health outcomes. These promising outcomes suggest that sustained investment in digital health care can help scale effective practices at the community level. Investing in digital health to support eCHIS adoption not only strengthens Ethiopia's primary health care model but also provides a valuable framework for similar low-resource settings aiming to enhance healthcare efficiency and reach. Further research is required to examine implementation bottlenecks of eCHIS to inform context-specific scale-up strategies and assess its cost-effectiveness and long-term impact in transforming the HEP to improve primary health care outcomes, efficiency, and service quality.

## Supporting information

**S1 File.** Survey questionnaire.xlsx.This file contains the survey questionnaires used to collect information from study participants. The first sheet provides variable definitions (data dictionary) in English and local languages (Amharic, Afaan Oromo, and Af-Somali), while the second sheet contains the answer choices for each variable.
(XLSX)

**S2 File.** Survey dataset.csv.This file contains the survey data with variables and their values used for analysis.
(CSV)

## Acknowledgments

We extend our gratitude to the Ministry of Health, woreda health offices, health center staff, and HEWs at the eCHIS project implementation sites for their essential engagement. We acknowledge the interviewers and supervisors for their hard work, dedication, and timely completion of the fieldwork. Finally, we are grateful to all study participants who took the time to respond to the survey questionnaires and provided invaluable information.

**Consent for publication:** Not applicable as all figures were developed as part of the work.

## Author contributions

**Conceptualization:** Gizachew Tadele Tiruneh, Girma Tadesse, Yared Kifle, Nebreed Fesseha, Biruk Bogale, Gete Mekuria, Dessalew Emaway.

**Data curation:** Gizachew Tadele Tiruneh, Tsegaye Shewangzaw, Biruk Bogale.

**Formal analysis:** Gizachew Tadele Tiruneh, Yared Kifle, Tsegaye Shewangzaw, Bezawit Mesfin Hunegnaw.

**Funding acquisition:** Gizachew Tadele Tiruneh, Girma Tadesse, Nebreed Fesseha, Gete Mekuria.

**Investigation:** Gizachew Tadele Tiruneh, Alemnesh Mirkuzie Hailemariam.

**Methodology:** Gizachew Tadele Tiruneh, Alemnesh Mirkuzie Hailemariam, Yared Kifle, Bezawit Mesfin Hunegnaw, Getnet Alem, Biruk Bogale, Dessalew Emaway.

**Project administration:** Gizachew Tadele Tiruneh, Girma Tadesse, Yared Kifle, Tsegaye Shewangzaw, Bezawit Mesfin Hunegnaw, Getnet Alem, Nebreed Fesseha, Gete Mekuria, Alemayehu Hunduma, Dessalew Emaway.

**Resources:** Dessalew Emaway.

**Supervision:** Gizachew Tadele Tiruneh, Girma Tadesse, Alemnesh Mirkuzie Hailemariam, Getnet Alem, Nebreed Fesseha, Gete Mekuria, Alemayehu Hunduma, Dessalew Emaway.

**Visualization:** Gizachew Tadele Tiruneh.

**Writing – original draft:** Gizachew Tadele Tiruneh.

**Writing – review & editing:** Gizachew Tadele Tiruneh, Girma Tadesse, Alemnesh Mirkuzie Hailemariam, Yared Kifle, Tsegaye Shewangzaw, Bezawit Mesfin Hunegnaw, Getnet Alem, Nebreed Fesseha, Biruk Bogale, Gete Mekuria, Alemayehu Hunduma, Dessalew Emaway.

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
