## [Decision Letter · Decision Letter 0]

19 Apr 2026

PONE-D-25-50830Effect of digitizing Community Health Information System on the delivery and use of maternal and child health services: Propensity score matching analysisPLOS One

Dear Dr. Tiruneh,

Thank you for submitting your manuscript to PLOS ONE. After careful consideration, we feel that it has merit but does not fully meet PLOS ONE’s publication criteria as it currently stands. Therefore, we invite you to submit a revised version of the manuscript that addresses the points raised during the review process.

A letter that responds to each point raised by the academic editor and reviewer(s). You should upload this letter as a separate file labeled ’Response to Reviewers’.A marked-up copy of your manuscript that highlights changes made to the original version. You should upload this as a separate file labeled ’Revised Manuscript with Track Changes’.An unmarked version of your revised paper without tracked changes. You should upload this as a separate file labeled ’Manuscript’.

We look forward to receiving your revised manuscript.

Kind regards,

Seyed Shahmy

Academic Editor

PLOS One

1. Please ensure that your manuscript meets PLOS ONE’s style requirements, including those for file naming. The PLOS ONE style templates can be found at

“The SSD and eCHIS projects are funded by the BMGF and CIFF, respectively, and implemented through JSI. The funders have no role in the interpretation and implications of the content in this paper. This responsibility lies solely with the authors.”

“The SSD and eCHIS projects are funded by the BMGF and CIFF, respectively, and implemented through JSI. The funders have no role in the interpretation and implications of the content in this paper. This responsibility lies solely with the authors.”

Reviewers’ comments:

Reviewer’s Responses to Questions

**Comments to the Author**

1. Is the manuscript technically sound, and do the data support the conclusions?

Reviewer #1: Yes

Reviewer #2: Yes

2. Has the statistical analysis been performed appropriately and rigorously? 

Reviewer #1: Yes

Reviewer #2: Yes

3. Have the authors made all data underlying the findings in their manuscript fully available?

Reviewer #1: Yes

Reviewer #2: Yes

4. Is the manuscript presented in an intelligible fashion and written in standard English?

Reviewer #1: Yes

Reviewer #2: Yes

5. Review Comments to the Author

Reviewer #1: Background:

Ensure all claims are properly referenced. For example, line 79 mentions.

Methods:

Highlight the imbalance and provide explanation whether weighting was applied during the analysis

Ensure consistency in describing age ranges: women of reproductive age (15–49 years)” should be explicitly linked to MNH and family planning data collection.

Ensure S1 File is properly cited and that the text indicates what S1 File contains

Measurement:

Use consistent past tense for methods

Table 1: Consider clarifying in text that the denominator differs per indicator and link this explicitly to survey design.

Explain how the first component was used to rank households

Add a short justification for why two out of three criteria was chosen, possibly referencing a previous study or standard.

Analysis:

Clarify why kebele-level PS were used, not individual-level, to help readers understand clustering at community level.

Stepwise-backward selection: mention that this was done separately for each subgroup, which you do, but could be clearer in flow.

Include a flowchart of sampling and survey process.

Discussion:

Several points are repeated: eCHIS impact on tracking, defaulter tracing, and workflow efficiency appears multiple times (lines 240–256, lines 257–266).

References to mHealth interventions improving maternal and child health outcomes (lines 263–266) partially overlap with earlier discussion (lines 257–262)

Sentences (lines 279-289) could be split for an easier read

While the discussion references prior studies, it could emphasize which results are consistent vs. which are novel in the Ethiopian context.

Some sentences describe improvements, but it would be stronger to directly link each improved outcome to the eCHIS feature responsible

Conclusion:

some sentences are redundant (lines 291–294 vs. 295–298). Condensing these would make the section more concise.

What type of investment or next steps are more critical and why?

Future research focus could be more precise.

Reviewer #2: the research article titled "Effect of digitizing Community Health Information System on the delivery and use of

"Maternal and child health services: Propensity score matching analysis" is a very interesting and urgent issue in today’s world. And hence, it fulfills the minimum standard for publication in the PLOS ONE journal. All the statistical analyses are very sound, and the findings revealed the current situations regarding maternal and child health. And finally, the conclusion and the findings are constructive enough.

6. PLOS authors have the option to publish the peer review history of their article (what does this mean?). If published, this will include your full peer review and any attached files.

Reviewer #1: No

Reviewer #2: No

---

## [Author Response · Author response to Decision Letter 1]

28 Apr 2026

A point-by-point response to reviewers and editor

PONE-D-25-50830: Effect of digitizing Community Health Information System on the delivery and use of maternal and child health services: Propensity score matching analysis

Dear Editor,

We, the authors, would like to thank the reviewer for their valuable comments. Our point-by-point responses to the reviewer and editor are below each comment. We also reviewed to ensure that this manuscript version conforms to the journal style.

1. Please ensure that your manuscript meets PLOS ONE’s style requirements, including those for file naming. The PLOS ONE style templates can be found at

“The SSD and eCHIS projects are funded by the BMGF and CIFF, respectively, and implemented through JSI. The funders have no role in the interpretation and implications of the content in this paper. This responsibility lies solely with the authors.”

Response: We have removed all funding-related statements from the Acknowledgments section in accordance with journal requirements. The revised Acknowledgments now read:

“We extend our gratitude to the Ministry of Health, woreda health offices, health center staff, and HEWs at the eCHIS project implementation sites for their essential engagement. We acknowledge the interviewers and supervisors for their hard work, dedication, and timely completion of the fieldwork. Finally, we are deeply grateful to all study participants who took the time to respond to the survey questionnaires and provided invaluable information.”

“The SSD and eCHIS projects are funded by the BMGF and CIFF, respectively, and implemented through JSI. The funders have no role in the interpretation and implications of the content in this paper. This responsibility lies solely with the authors.”

Response: We confirm that the Funding Statement remains unchanged and is included only in the designated section of the submission form:

“The SSD and eCHIS projects are funded by the Gates Foundation and the CIFF, respectively, and implemented through JSI. The funders had no role in the interpretation of the findings or the conclusions presented in this manuscript. The authors bear sole responsibility for the content.”

Response: The amended statements are included in the cover letter.

Response: We have carefully reviewed and updated the reference list to ensure completeness, accuracy, and consistency with journal guidelines. No retracted articles were cited. Any minor corrections have been incorporated and are reflected in the revised manuscript.

Reviewers’ comments:

Review Comments to the Author

Reviewer #1: Background:

Ensure all claims are properly referenced. For example, line 79 mentions.

Response: Thank you. We have reviewed the Background section and added appropriate references to support all key claims.

Methods:

Highlight the imbalance and provide explanation whether weighting was applied during the analysis

Response: Thank you for this important point. We have clarified this in the Discussion and Limitations sections. Specifically, we now note that residual imbalance may persist due to the study design and that matching could not fully address differences between groups. The revised text reads:

“Additionally, because the study was not originally designed to evaluate eCHIS, baseline imbalances between intervention and comparison groups may persist despite matching. These likely reflect differences in geographic, socioeconomic, and health system contexts across clusters, which could influence both exposure and outcomes. Residual confounding may therefore affect the estimated effects.” Page 19, lines 323-27 of the revised manuscript with track changes

Ensure consistency in describing age ranges: women of reproductive age (15–49 years)” should be explicitly linked to MNH and family planning data collection.

Response: Thanks for the feedback. This has been revised for clarity and consistency. The text now reads:

“The SSD baseline survey used a stratified, multistage sampling approach to recruit three groups of women aged 15–49 years: 3,611 women of reproductive age, 1,922 women with infants aged 0–11 months, and 1,368 women with children aged 12–23 months.” Page 7, lines 115-17 of the revised manuscript with track changes

Ensure S1 File is properly cited and that the text indicates what S1 File contains

Response: Noted. The S1 File is now properly cited in the text, and its contents are clearly described.

Measurement:

Use consistent past tense for methods

Response: Thank you. We have revised the Methods section to ensure consistent use of past tense throughout. Page 9, line 149 of the revised manuscript with track changes

Table 1: Consider clarifying in text that the denominator differs per indicator and link this explicitly to survey design.

Response: We have clarified this in the manuscript. The revised text specifies that:

“The dependent variable of interest was the RMNCH service uptake, which was expected to be affected by the eCHIS intervention and was measured by the household surveys among three groups of women aged 15–49 years: women of reproductive age for family planning indicators, women with infants aged 0–11 months for MNH indicators, and women with children aged 12–23 months for immunization indicators.” Page 8-9, line 143-46 of the revised manuscript with track changes

Explain how the first component was used to rank households

Response: We have clarified this as follows:

“The first principal component, which captured the largest variation in asset ownership and living conditions and had the highest eigenvalue, was retained to represent underlying socioeconomic status and used to generate a continuous wealth score for each household.” Page 10, line 159-62 of the revised manuscript with track changes

Add a short justification for why two out of three criteria was chosen, possibly referencing a previous study or standard.

Response: Thank you for this important question. The decision to classify women as having autonomy when they met at least two of the three criteria was made to ensure that the measure reflects a minimum consistent level of decision-making power across multiple domains, rather than isolated or context-specific participation in a single domain. Using a two-out-of-three threshold reduces the likelihood of misclassifying women as autonomous based on a single favorable response and provides a more robust proxy of sustained autonomy in household decision-making. This approach is consistent with practices in demographic and health research where composite empowerment indicators are constructed to capture multidimensional constructs rather than single-item responses.

Analysis:

Clarify why kebele-level PS were used, not individual-level, to help readers understand clustering at community level.

Response: We have clarified this in the Methods section:

“Propensity scores were calculated for each kebele to estimate adjusted intervention effects, using a logit model for the three groups of mothers separately, accounting for the clustered sampling design in which households within the same kebele share similar contextual characteristics and reducing bias from correlated exposures.” Page 11, lines 172-75 of the revised manuscript with track changes

Stepwise-backward selection: mention that this was done separately for each subgroup, which you do, but could be clearer in flow.

Response: Thank you. We have clarified that this procedure was conducted separately for each subgroup to improve readability and transparency.

Include a flowchart of sampling and survey process.

Response: We appreciate the suggestion. A flowchart illustrating the sampling and survey process has been added as Figure 1. Page 7, lines 131 of the revised manuscript with track changes

Discussion:

Several points are repeated: eCHIS impact on tracking, defaulter tracing, and workflow efficiency appears multiple times (lines 240–256, lines 257–266).

Response: Thank you. The Discussion has been revised to remove redundancies and streamline overlapping content.

References to mHealth interventions improving maternal and child health outcomes (lines 263–266) partially overlap with earlier discussion (lines 257–262)

Response: We have consolidated overlapping references to mHealth evidence into a single coherent paragraph to improve clarity.

Sentences (lines 279-289) could be split for an easier read

Response: These sentences have been revised and split to improve readability.

While the discussion references prior studies, it could emphasize which results are consistent vs. which are novel in the Ethiopian context.

Response: We have strengthened the Discussion to explicitly distinguish findings consistent with prior literature from those that are novel in the Ethiopian context.

Some sentences describe improvements, but it would be stronger to directly link each improved outcome to the eCHIS feature responsible

Response: Thank you for these constructive comments. We have comprehensively revised the Discussion section to improve clarity, coherence, and analytical depth. Specifically, we removed repetitive statements and consolidated overlapping discussions of mHealth evidence into a single coherent paragraph. In addition, we strengthened the Discussion by clearly distinguishing findings that are consistent with existing literature from those that are novel to the Ethiopian context. Page 16-19, lines 257-31 of the revised manuscript with track changes

Conclusion:

some sentences are redundant (lines 291–294 vs. 295–298). Condensing these would make the section more concise.

Response: Thank you. The Conclusion has been condensed to remove repetition and improve clarity. Page 20, lines 335-37 of the revised manuscript with track changes

What type of investment or next steps are more critical and why?

Response: We have expanded this section as follows:

“However, maximizing its impact will require addressing implementation bottlenecks, strengthening infrastructure, and aligning with service delivery policies.” Page 20, lines 310-11 of the revised manuscript with track changes

Future research focus could be more precise.

Response: We have refined this to state:

“Further research is required to examine implementation bottlenecks of eCHIS to inform context-specific scale-up strategies and assess its showcase eCHIS’s cost-effectiveness and long-term impacts in transforming the HEP to improve primary health care outcomes, efficiency, and service quality.” Page 20, lines 341-42 of the revised manuscript with track changes

Reviewer #2: the research article titled "Effect of digitizing Community Health Information System on the delivery and use of

"Maternal and child health services: Propensity score matching analysis" is a very interesting and urgent issue in today’s world. And hence, it fulfills the minimum standard for publication in the PLOS ONE journal. All the statistical analyses are very sound, and the findings revealed the current situations regarding maternal and child health. And finally, the conclusion and the findings are constructive enough.

Response: We sincerely thank the reviewer for the positive and encouraging feedback. We appreciate the recognition of the study’s relevance, methodological rigor, and contribution to understanding maternal and child health service delivery.

---

## [Editor Report · Decision Letter 1]

12 May 2026

Effect of digitizing Community Health Information System on the delivery and use of maternal and child health services: Propensity score matching analysis

PONE-D-25-50830R1

Dear Authors,

We’re pleased to inform you that your manuscript has been judged scientifically suitable for publication and will be formally accepted for publication once it meets all outstanding technical requirements.

An invoice will be generated when your article is formally accepted. Please note, if your institution has a publishing partnership with PLOS and your article meets the relevant criteria, all or part of your publication costs will be covered. Please make sure your user information is up-to-date by logging into Editorial Manager at Editorial Manager® and clicking the ‘Update My Information’ link at the top of the page. For questions related to billing, please contact billing support.

Kind regards,

Seyed Shahmy

Academic Editor

PLOS One

Additional Editor Comments (optional):

Good work and the reviewer comments were addressed appropriately.

Reviewers’ comments:

---

## [Editor Report · Acceptance letter]

PONE-D-25-50830R1

PLOS One

Dear Dr. Tiruneh,

I’m pleased to inform you that your manuscript has been deemed suitable for publication in PLOS One. Congratulations! Your manuscript is now being handed over to our production team.

Lastly, if your institution or institutions have a press office, please let them know about your upcoming paper now to help maximize its impact. If they’ll be preparing press materials, please inform our press team within the next 48 hours. Your manuscript will remain under strict press embargo until 2 pm Eastern Time on the date of publication. For more information, please contact onepress@plos.org.

Kind regards,

on behalf of

Dr. Seyed Shahmy

Academic Editor

PLOS One